# Validity and reliability of a physical literacy knowledge, attitudes, self-efficacy and behaviors questionnaire for early childhood educators (PLKASB-ECE): An exploratory factor analysis

**Lynne M. Z. Lafave**[1,2]*, **Nadine Van Wyk**[2], **Alexis D. Webster**[1], **Joyce Hayek**[1], **Mark R. Lafave**[2]

**1** Department Health and Physical Education, Department Health and Physical Education, CHEERS Lab, Mount Royal University, Calgary, Alberta, Canada, **2** Department Health and Physical Education, Mount Royal University, Calgary, Alberta, Canada

* llafave@mtroyal.ca

## Abstract

### Background

Nurturing physical literacy in young children offers a unique opportunity to address global physical inactivity trend. Early childhood education and care (ECEC) environments, with their extensive reach into this age group, and early childhood educators, through their daily interactions with children, are strategically positioned to influence children's physical literacy development. However, enhancing educators' ability to foster physical literacy requires valid and reliable assessment tools to measure holistic physical literacy constructs (i.e., cognitive, affective, behavioral) to assess the impact of physical literacy educational interventions.

### Objective

The purpose of this study was to develop a holistic digital tool to measure physical literacy knowledge, attitudes, self-efficacy, and behaviors of early childhood educators (PLKASB-ECE) for both their professional teaching context as well as the educator's own personal physical literacy behaviours.

### Methods

This study was conducted in two phases. Phase 1 involved item generation and content validation. Phase 2 employed a cross-sectional validation study design to assess the psychometric properties of the PLKASB-ECE tool.

### Results

Based on a literature review an initial 19-item instrument was developed that subsequently underwent three rounds of expert content validation. Six additional items were added,

**Data Availability Statement:** The dataset used for the present study (inclusive of the minimal dataset

necessary to replicate the findings presented) cannot be deposited in a public data repository for ethical reasons. Study participants did not provide written consent for their data to be made publicly available. Article 3.13 of TCPS 2 informs researchers that data sharing for peer review does not require participant consent whereas the storage and future unspecified use of human data requires participant consent [Chapter 3]. Participants were informed their data would be used for this study, but no consent was provided for their data to be made publicly available. Any requests for access to any data related the present study should be addressed to Lynne Lafave llafave@mtroyal.ca, or the Research Compliance Officer at Mount Royal University via hreb@mtroyal.ca, who oversee ethical conduct of research practices.

**Funding:** This research was funded by Government of Alberta, Children Services (#ACS565189) and LL received this award. https://www.alberta.ca/children-and-family-services. The funder had no role in the study design, data collection and analysis, decision to publish, or preparation of the manuscript.

**Competing interests:** The authors have declared that no competing interests exist.

resulting in a final 25-item self-report measure with a readability score equivalent to an 8[th] grade reading level (Flesch-Kincaid Grade Level: 8.7). This included 1 global rating item, 2 qualitative response items, 7 knowledge items, and 15 items addressing physical literacy attitudes, self-efficacy and behaviors using a 7-point Likert response scale. The PLKASB-ECE tool was administered to 470 educators in Alberta, Canada between 2019 to 2022. The 15 items were subjected to exploratory factor analysis and resulted in a five-factor scale with one item not loading. The five-factor scale held with the final 14 items with loadings ranging from 0.481 to 0.886, Cronbach's alpha ranging from 0.70 to 0.82, with ordinal omega ranging from 0.72 to 0.82. Usability, as assessed by completion time, was 8.15 minutes.

## Conclusion

These findings demonstrate good indices of reliability and validity for the PLKASB-ECE tool. This tool will be valuable as a research outcome measure for assessing educational interventions aimed at enhancing educators' understanding, confidence, and strategies for fostering holistic concepts of physical literacy in young children within ECECs.

## Introduction

Physical activity, bodily movement resulting from skeletal muscle activation leading to increased energy expenditure, has been identified as a protective factor for human health [1]. Early childhood physical activity has been favorably associated with child health indicators such as body type, gross motor skills, fine motor skills, cardiometabolic health, cognitive development and other psychosocial factors like self-efficacy, social functioning, executive functioning and language development [2]. Affinity and the regular engagement in physical activity have foundational origins in early childhood that have been shown to track into adolescence and adulthood [3–5]. The benefits of physical activity throughout the lifespan are abundant including reduced risk of death from non-communicable diseases (NCDs) such as cancer, heart disease, stroke, diabetes, and improvements to mental health and wellness [6]. In contrast, physical inactivity, the absence of sufficient physical activity required to meet the current physical activity recommendations, has been identified as a leading risk factor for premature mortality from NCDs [7].

Globally, there is a strong movement to build healthy communities and individual capacity in the realm of physical activity in an effort to value health as a universal right [1]. The United Nations General Assembly adopted a special resolution to guide policy development to increase physical activity internationally by 2030 [1]. In Canada, a number of guiding policies have been initiated to encourage and prioritize physical activity such as the Canadian 24-hour movement guidelines (3) and the Canadian Sport Policy [8]. The WHO *Global Action Plan on Physical Activity 2018–2030* set a target of a 15% relative reduction in global physical inactivity prevalence for adults and adolescents by 2030. However, global progress on reducing physical inactivity has been slow despite these efforts with surveillance revealing 27% of adults and 80% of adolescents worldwide are insufficiently active [9,10].

One group often overlooked has been early childhood aged 0–5 years, which creates a challenge considering the scope of this problem is well documented for most of society [2,11]. No target for reduction in physical inactivity nor surveillance for this age group are included in these global documents and almost no data for this age group is available [11]. The Canadian

Health Measures Survey data [12] provides limited data suggesting that only two thirds of Canadian preschoolers (3-4-year-olds) meet the physical activity 24-Hour Movement Guidelines (≥180 minutes/day of total physical activity, including ≥60 minutes/day of energetic play). A systematic review of objectively measured physical activity in center-based child care revealed that physical inactivity levels are high across worldwide geographical locations [13].

Physical inactivity in the early years has direct linkages to future adult health outcomes. Research indicates movement habits formed in the early years often track into adulthood [4] making early childhood an opportune time to cultivate the foundation of physical literacy. Statistics from OCED countries indicate that, on average, approximately 77% of children 3 to 5 years are enrolled in some type of early childhood education and care (ECEC) environment [14]. In 2023, over 56% of Canadian children, between the ages of 0–5 years, were registered in licensed or unlicensed child care centers [15]. These settings provide a fertile ground for targeting strategies for nurturing physical literacy in the early years.

A key strategy to implement physical activity recommendations in young children is through a physical literacy lens. Physical literacy encompasses a comprehensive approach to increasing physical activity participation which in turn supports child development [16]. Canada's Physical Literacy Consensus Statement defines physical literacy as *"the motivation, confidence, physical competence, knowledge, and understanding to value and take responsibility for engagement in physical activities throughout life" and is considered distinct* from physical activity since this exclusively relates to bodily movement [17]. Physical literacy addresses the lifelong journey of who, what, where, when, why, and how to cultivate the engagement in physical activity that enhances personal health. Physical literacy can be conceptualized in a framework of four interrelated domains: affective, physical, cognitive, and behavioral [17]. This holistic approach emphasizes the social processes associated with lifelong learning and sustained physical activity engagement [17–19]. Physical literacy experts posit that a strong physical literacy foundation will improve an individual's overall health and wellness through the building of an individual's positive relationship with daily movement [19–22].

High quality child care has long-term effects on a child's cognitive, academic, and language development tracking into adolescence [23]. Early childhood educators' personal health knowledge and practices contribute to high quality care and are vital in creating a health-supportive environment, influencing a child's lifelong relationship with physical activity [24,25]. However, there is considerable variability in early childhood educator professional preparation (education level) where some educators have completed an early childhood education certificate after completing high school whereas others report having college degrees with early childhood education diplomas. In a meta-analysis, investigators found a strong positive predictive relationship between professional preparation and quality early learning education experiences [26]. This connection tracks into health environments where educators with some college education demonstrate better adherence to physical activity best practices within ECEC environments compared to educators attaining high school qualifications [27]. In addition, educators who express value towards physical literacy and engage in higher levels of personal physical activity provide more physical activity opportunities and engage in role modeling with children in their care [28–31]. In the context of physical literacy and early childhood education, research suggests that educator's knowledge, practices, and efficacy beliefs influences their ability and intention to facilitate movement for children [32,33].

A large portion of Canadian children spend 30 hours or more in ECEC arrangements [15]. As a result, educators are increasingly recognized as 'agents of change' tasked with fostering children's physical literacy development [29,30,34]. Barratt et al., [34] proposed a conceptual framework that suggests an effective early childhood physical literacy pedagogue emerges when educator physical literacy capabilities are merged with educator identity and play-based

pedagogy. Despite this critical role, no early childhood education certificate or degree programs in Canada offer individual physical literacy or motor development courses [35]. Educators may value the broad concept of movement, but there are gaps in *'knowledge of'* and *'practical skills for'* the holistic philosophy and implementation of physical literacy instruction within ECEC [24,34,35]. Offering targeted physical literacy-based professional development to educators has been proposed as one strategy to fostering higher-quality movement experiences for children attending ECEC [34]. However, there are limited assessment tools with reported validity and reliability measures available to evaluate early childhood educators' support of young children's physical activity [33] and no comprehensive physical literacy tools [36] for use in intervention studies. The early childhood physical literacy pedagogue framework posits that effectiveness manifests in the three physical literacy constructs: knowledge, perceived capability, and practices. In addition, considering that educators' personal physical activity engagement correlates with increased physical activity curriculum offering in their professional practice [28–31], assessing educators' personal physical literacy may provide additional insights into effectiveness. Having a valid and reliable comprehensive assessment tool could provide an understanding of educators' capacity to implement physical literacy education within their professional practice. The purpose of this study was to develop a holistic tool to measure early childhood educator knowledge, attitudes, self-efficacy, and behaviors of physical literacy for both their professional teaching context as well as educator's own personal context.

## Methods

This study was conducted in two phases: Phase 1 involved the development of tool items and content validation followed by Phase 2 which involved structural validity and reliability evaluation.

### Phase 1 PLKASB-ECE item generation and content validation

**Item generation.** We considered physical literacy within the holistic framework of (i) physical activity behavior, (ii) attitude towards a physically active lifestyle, (iii) motivation as well as, (iv) knowledge and (v) self-efficacy towards physical activity [16]. First, we developed a pool of items based on a literature review. A search of SPORTDiscus, MEDLINE, and CINAHL was conducted to identify concepts of early childhood educator's knowledge, attitude, self-efficacy, and behaviors related to physical literacy in childcare environments. Keywords, along with truncated versions, included "physical literacy", with measurement, assessment, monitoring, evaluation, questionnaire, survey, knowledge, attitudes, early childhood education, early childhood educator, daycare, childcare, and long-term daycare. Reference lists of articles meeting the search strategy and relevant to physical literacy and measurement were checked for additional related works. Decisions regarding item generation were guided by the Critical Ecology Framework and Social Cognitive Theory [37,38].

**Content validity.** A systematic approach was used for scale development [39]. The content validation process followed a modified Ebel procedure with a panel of three experts (two early childhood physical literacy and one health promotion expert) examining the initial items for breadth and scope of concepts addressed. Each item was assessed for importance and clarity. Quantitative feedback was collected on item importance rating using a four-point Likert scale (1 = not relevant, 2 = somewhat relevant, 3 = quite relevant, 4 = highly relevant). The item content validity index (I-CVI) was calculated on item relevance by dividing the number of experts issuing a judgment of "3 or 4" divided by the total number of experts [40]. Qualitative feedback was collected on the clarity of each item (item included 'as written' or 'rewording required') as well as comments for missing items. Items with an I-CVI greater than 0.78 were

included in the final tool and those less than 0.78 guided discussion decisions about item revision, rejection, or the development of new items. Items revised or newly created were then subject to expert review in subsequent rounds.

**Face validity.** To address face validity of the scale, two in-service educators, two pre-service educator students, and two educator university faculty members were identified for their expertise to provide a stakeholder perspective. The survey questions were provided to individuals to read and the research team took notes of verbal feedback regarding item clarity, comprehension, and suitability to the community.

**Readability analysis.** Readability was assessed using the Flesch-Kincaid Grade Level (F-K) and Flesch Reading Ease (FRE) for readability. These formulas consider factors such as the number of words per sentence, syllables per word, vocabulary, and word complexity. The F-K score was the primary outcome in evaluating readability [41]. The F-K score has a ceiling effect of 12 and is valid for written text between Grade 5 and college level [42]. The readability scores were obtained using Microsoft Word software (MS Office for PC, Microsoft, Inc. 2007).

## Phase 2 structural validity and reliability assessment of PLKASB-ECE scale

A cross-sectional validation study design was implemented to assess the psychometric properties of the PLKASB-ECE tool.

**Participants.** Participants were sourced from a larger research project *"CHEERS HEAPful of FUN*: *raising healthy Albertans"* [43]. This study examines an intervention aimed at supporting early childhood educators to promote best practices for eating and physical activity within an early childhood education and care environments serving children ages 2–5 years. ECEC centers were randomly selected for recruitment from 02/08/2019 to 31/08/2022 using postal codes to stratify the selection of ECEC centers from large urban population centers (population > 100,000), medium population centers (30,000–99,000), small population centers (1000–29,999), and rural areas (population < 1000) throughout the province [44]. Large urban population centers represent approximately 70% of the provinces population with 16% and 13% falling in the medium and small population centers, respectively [45]. We looked to balance the proportion of center recruitment relative to the overall provincial population. In the first year of data collection, center directors were invited to participate through phone correspondence, provided with general information on the study and allocated to either intervention or control group based on center directors' assessment of both available time and capacity. Centers that agreed to participate invited three educators from their center to join the research study. Each educator received a personal email containing: information about study participation, private secure digital access to their consent form, and contact information for a trained research associate to answer potential questions. Inclusion criteria include: 1) licensed facility-based centers 2) providing care for a minimum of 15 children aged 2 to 5 years, 3) access to a computer and internet connection, and 4) not currently enrolled in any other intervention to improve nutrition and activity practices. Exclusion criteria include unlicensed ECEC centers, and family day home or after school care program. Each participating director was requested to identify three staff members (classroom educator or director) to be included in the study, with a preference to include two educators and the center director.

**Online interface.** The questionnaire was built for digital delivery administration in Qualtrics® online-based survey platform. This survey system is compatible across computer, tablet, and mobile smartphone platforms (e.g., Apple, Safari, Google Chrome, Internet Explorer, Mozilla Firefox). Qualtrics® facilitates the optimization of delivery to match the user's interface by adjusting the screen size and formatting. The digital PLKASB-ECE questionnaire is presented as three questions per page with a progress tracking bar at the top of the survey. The

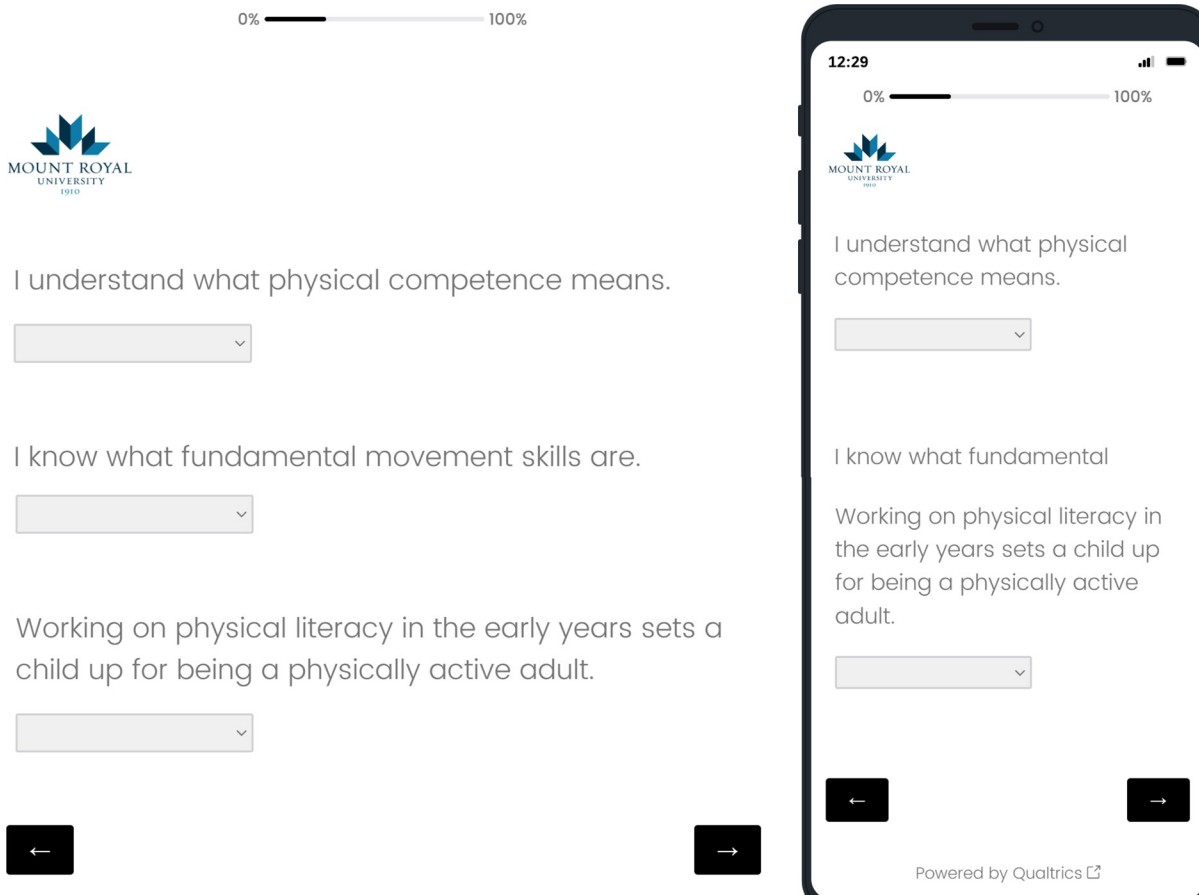

**Fig 1. Screenshot of questionnaire in digital platform for web or smartphone.**

platform also records the duration (time to complete the survey). A sample screenshot of the interface for the survey tool is presented in Fig 1. Once participant eligibility was verified and consent was provided, unique participant identifiers were loaded to the system. Participants received an email link to the survey and were required to enter name and password to access the survey.

**Sample size considerations.** In exploratory factor analysis, a suggested minimum subject-to-item (N:p) ratio ranges from 5:1 to 10:1 [46,47]. Goretzko and colleagues indicate that a target 400 participants or more should be sought for exploratory factor analysis to get reliable factor patterns and scores [48].

**Statistical analysis.** Data analysis for the exploratory factor analysis, Cronbach's alpha, and ordinal omega were conducted using statistical software R (version 4.2.2). Descriptive statistics were performed using SPSS statistical package version 26 (SPSS Inc, Chicago, IL). Utility was evaluated by documenting the length of time taken to complete the PLKASB-ECE survey as recorded in the survey platform housed in Qualtrics® [49].

**Exploratory factor analysis.** To investigate construct validity, we conducted an exploratory factor analysis (EFA) with an oblimin rotation using the maximum likelihood method to evaluate the internal structure of the 15 items plus the knowledge sum score from the PLKASB-ECE. The global rating scale was not included in the EFA. The first step was to determine whether an EFA was appropriate for the data by examining the Kaiser-Meyer-Olkin (KMO)

and Bartlett's Test of Sphericity, where appropriateness is indicated by KMO minimum values of greater than 0.6 and significant Bartlett's tests [50]. Then, we inspected eigenvalues, scree plots, and parallel analysis (PA), to decide the number of factors to extract. We then used maximum likelihood method to extract the factors and an oblimin rotation method to compare two different models. We examined both models' items loadings onto factors in coherence with theory and the following fit indices were used to evaluate model fit: Bayesian Information Criterion (BIC), Tucker-Lewis index (TLI), and root-mean-square error of approximation (RMSEA). The BIC is a relative fit index where smaller values relative to the other comparative models indicate better fit. TLI values greater than .95 suggest good fit [51]. Finally, RMSEA estimates of $<0.01$, $< .05$, and $< .08$ denote excellent, good, and moderate fit, respectively [52].

**Internal consistency.**   Internal consistency estimates were conducted in order to determine the reliability of the PLKASB-ECE. Cronbach's alpha and ordinal omega were used to estimate internal consistency reliability coefficients for each subdomain in the best-fitting model [53], as well as for the combination of all items. Ordinal omega was included as an assessment as it can provide more accurate estimates for multidimensional and ordinal data [53].

**Ethics statement.**   This study followed the Tri-Council Policy Statement: Ethical Conduct for Research Involving Humans (TCPS 2) and the Mount Royal University Human Research Ethics Board approved the study protocol (no. 101768). All participants were given ample time to read and review the consent forms prior to participating in the research study.

## Results

### Phase 1. PLKASB-ECE item generation and content validation

**Item generation.**   The structured literature search revealed 106 articles. After removing duplicates using the related keywords from our search strategy, 24 articles were relevant to physical literacy and measurement. From these results, no specific tool measuring early childhood educator knowledge, attitudes, self-efficacy, and behaviors of physical literacy within their teaching context was identified. Four articles included assessment concepts related to preschool aged children (3–5 years) but focused solely on motor skill assessment aspects of physical literacy [54–57]. Six review articles [58–63] examined literature on physical literacy. Twelve articles focused on the measurement of motor skills related to physical literacy in school-aged children, specifically from kindergarten to grade 6 [64–71] and grades 7 to 12 [18,72–74]. Additionally, one article described the development of a physical literacy questionnaire for primary school teachers [75], and another provided a qualitative assessment of physical literacy among this group [76]. Using a holistic framework of physical literacy, all articles were combed for items addressing knowledge, attitudes, self-efficacy, and behaviors of physical literacy to be adapted to an early childhood educator context. A pool of nineteen questions were adapted for this study.

**Content validity.**   After expert review, 6 items were considered to have insufficient content validity (CVI $<$ 0.78 for importance) and were brought to the face-to-face discussion. The remaining 13 items had sufficient content validity to be included in the final scale with eight of the items flagged for rewording which were also brought to the face-to-face discussion. The 6 items not reaching consensus were deemed appropriate for inclusion (I-CVI$>$.78) after discussion, rewording, and revisions for clarity. Specifically, the term "I plan lots" was rephrased to "I plan 2" to reduce subjective interpretation of 'lots' and reduce the cognitive load in responding to questions. The knowledge question "*Motivation, confidence, knowledge and physical competence are all important in developing physical literacy*" was broken down into individual

items for respondent clarity and specificity which resulted in 3 additional knowledge items. Three additional items were added through discussion that addressed perceived benefits of physical literacy education for children, breaking indoor and outdoor activities into two questions, and adding a second qualitative item asking for examples of how educators support children to be physically active in their center. This resulted in 25 items in the final questionnaire, see S1 Appendix for I-CVI table.

**Face validity.**   Suggested changes from in-service educators, undergraduate educator students and instructors from an early learning and child care university program included refinements in language level and word substitutions.

**Readability analysis.**   In the first version of the questionnaire, the Flesch-Kincaid Grade Level (F-K) score for each individual questionnaire item ranged from a Grade 5.2 to 11.5 reading level. After suggested changes from physical literacy experts during the face validity feedback, the research team revised wording which resulted in an F-K score of Grade 5.2 to 9.5 reading level for each item. The finalized questionnaire as a whole resulted in an F-K score of 8.7 Grade reading level, an FRE of 53.1 readability, and a 0% score for passive voice. See S2 Appendix for finalized items.

## Phase 2 structural validity and reliability assessment of PLKASB-ECE scale

**Participants.**   Demographic characteristics of the educators and the child care centers where they work are presented in Table 1. A total of 470 educators participated in this study. On average 63.8% of ECEC centers were privately operated and 76.0% of these centers were from large urban population cities. The majority of the educators were women (97.7%) with an average age of 39.5 years. Approximately half (50.2) of the workforce had completed a two-year diploma in early childhood education followed by a university degree (23.8%) with the remaining completing the orientation course (14.0%) or one-year certificate (11.9%).

**Exploratory factor analysis—comparison of factor structures.**   Of the 25 items on the PLKASB–ECE, one is a global rating item (item #1), seven are knowledge items (items #2–8) scored correct or incorrect for a knowledge sum score, and two are qualitative questions (item #19 & #25). The remaining 15 items are scored on a scale of 1 to 7, with one scored 0 to 7 (#20). The global rating item and qualitative items were not included in the statistical analysis. The 15 items plus the knowledge sum score (16 items total) were subjected to exploratory factor analysis (EFA). Prior to performing EFA, the suitability of data for factor analysis was assessed. The KMO elicited high values for all items with the lowest being 0.76 for the knowledge composite score and the overall value was 0.86. Bartlett's test of sphericity was significant, $\chi^2 = 2745.94$, df = 120, p < 0.001. Both supporting the factorability of the correlation matrix.

To determine the number of factors, the eigenvalue greater than 1 rule suggested a four-factor solution. However, the results of PA, which showed five components with eigenvalues exceeding the corresponding criterion values for randomly generated data matrix of the same size. Therefore, we compared both of the four- and five-factor solutions. The fit indices indicate that the five-factor model better fit the data than the four-factor model, as shown in Table 2. Overall, we determined that the five-factor model would be the most ideal for the PLKASB-ECE survey. This decision was based on considerations of model interpretability, fitness indices, and alignment with theoretical frameworks.

In the five-factor model, all except two items loaded onto a factor, with loadings ranging from 0.481 to 0.886. The two items of the PLKASB-ECE survey that did not load adequately on any factors (factor loadings < 0.30) were a personal behavior item (item #24—*As a child, I was physically active outside of school time*) and the knowledge sum score. After removing these items from the model, the oblimin rotated solution revealed the presence of a simple

**Table 1. Characteristics of ECEC centers and ECEC educators.**

| Characteristics | Frequency (%) |
|---|---|
| **ECEC Center** | |
| Geographic location | |
| large urban population center | 357 (76.0) |
| medium population centers | 45 (9.6) |
| small population centers | 68 (14.5) |
| Auspice | |
| Not-for profit child care | 170 (36.2) |
| For-profit child care | 300 (63.8) |
| **ECEC Educator** | |
| **Highest education achieved** | |
| Level 1—Orientation Course | 66 (14.0) |
| Level 2–1-year certificate | 56 (11.9) |
| Level 3–2-year diploma | 236 (50.2) |
| University degree | 112 (23.8) |
| **Age (yrs ± SD)** | 39.5 ± 11.0 |
| **Gender** | |
| Men | 11 (2.3) |
| Women | 459 (97.7) |
| **Position** | |
| Director | 137 (29.1) |
| Educator | 289 (61.5) |
| Cook/Chef | 11 (2.3) |
| Owner/operator | 33 (7.0) |

ECEC, early childhood education and care center; Not-for profit, child care services organized and operated for a purpose other than profit; For-profit, surplus funds that exceed the cost of operation can be distributed among owners/shareholders; yrs, years; SD, standard deviation.

structures [77] with five components showing a number of strong loadings and variables loading substantially onto only one factor, as presented in Table 3. The five-factor solution explained a total of 58% of the variance, with each factor contributing between 13% and 11%.

The five-factor model used the oblimin rotation, thus allowing the factors to correlate. Factor correlations are presented in Table 4. The factors were weakly to moderately correlated, suggesting the factors represent related but distinct constructs.

**Internal consistency.** Measures of internal consistency are presented in Table 5. The 14 items of the PLKASB-ECE showed good reliability, $\alpha = 0.86$ and $\omega_{ordinal} = 0.84$, with all factors having acceptable ordinal omega and Cronbach's alpha values ($>0.7$) [50].

**Table 2. Examination of the model fit to data indices.**

| Fit indices | Four-Factor Model | Five-Factor Model |
|---|---|---|
| BIC | -156.72 | **-186.78** |
| TLI | 0.879 | **0.935** |
| RMSEA 90% CI (LL, UL) | 0.075 (.064, .086) | **0.055 (.043, .068)** |

BIC, Bayesian Information Criterion; TLI, Tucker-Lewis index; RMSEA, root-mean-square error of approximation; CI, confidence interval; LL, lower limit; UL, upper limit bolded values indicate better fit.

**Table 3. Item loadings for the five-factor model.**

| Item Question | Factor 1 | Factor 2 | Factor 3 | Factor 4 | Factor 5 | h2 |
|---|---|---|---|---|---|---|
| I understand what physical competence means. [#9] | 0.076 | **0.691** | 0.079 | 0.032 | 0.016 | 0.63 |
| I know what fundamental movement skills are. [#10] | -0.003 | **0.886** | 0.007 | 0.010 | -0.010 | 0.79 |
| I feel confident including movement skills in my daily practice. [#12] | 0.007 | 0.231 | **0.481** | 0.076 | 0.172 | 0.58 |
| I feel confident identifying when a child is struggling with a movement skill. [#13] | -0.037 | -0.018 | **0.769** | 0.002 | 0.041 | 0.57 |
| I feel confident giving children strategies to help them improve their movement skills. [#14] | 0.154 | 0.129 | **0.600** | 0.052 | -0.072 | 0.60 |
| I plan 2 or more INDOOR physical activity experiences each day for children at my center. [#15] | **0.734** | -0.026 | 0.116 | 0.006 | -0.042 | 0.61 |
| I plan 2 or more OUTDOOR physical activity experiences each day for children at my center. [#16] | **0.756** | 0.023 | -0.097 | -0.022 | 0.139 | 0.57 |
| I ask the children to tell me what they are learning when they engage in physical activity. [#17] | **0.603** | 0.127 | 0.048 | 0.144 | -0.120 | 0.54 |
| I do 30 minutes of heart pumping physical activity ___ days per week. [#20] | 0.020 | -0.019 | -0.051 | **0.601** | -0.031 | 0.33 |
| When planning my day I think about how I will find time to include physical activity for myself. [#21] | 0.009 | -0.007 | 0.012 | **0.882** | 0.004 | 0.79 |
| I enjoy being physically active. [#22] | -0.022 | 0.120 | 0.050 | **0.497** | 0.201 | 0.44 |
| Working on physical literacy in the early years sets a child up for being a physically active adult. [#11] | -0.039 | 0.295 | -0.046 | -0.088 | **0.524** | 0.38 |
| I support children to be physically active in our center. [#18] | 0.227 | -0.066 | 0.249 | -0.044 | **0.517** | 0.53 |
| Building my physical literacy skills can support my personal health throughout my life. [#23] | 0.009 | -0.016 | 0.017 | 0.091 | **0.835** | 0.75 |

Acceptable item loadings are bolded. $h^2$ = communality.

**Usability measure.** The median time taken to complete the 25 item PLKASB–ECE for all participants was 8.15 minutes (IQR 5.56–14.16) inclusive of responding to the two qualitative items.

**PLKASB-ECE scoring.** The quantitative component of PLKASB-ECE is a self-administered instrument that measures early childhood educators' knowledge, attitude, self-efficacy, and behaviors related to physical literacy in child care environments. The instrument consists of six subscales. The first is the Knowledge Sum Score (n = 7) measuring educators' physical literacy knowledge by summing all correct answers for the score. The second subscale is the Perception of Knowledge (n = 2) measuring educators' perception of their understanding related to physical literacy concepts. The third subscale is Attitude (n = 3) measuring educators' attitudes regarding physical literacy and health outcomes. The fourth subscale is Self-Efficacy (n = 3) measuring educator's self-efficacy towards developing physical literacy skills in the early learning environment. The fifth subscale is Environment Behaviors (n = 3) measuring educators' perception of their physical literacy promotion practices in the ECE environment. The sixth subscale is Personal Behaviors (n = 3) measuring perception of personal physical activity behaviors (Table 6). The qualitative component of the tool includes two open ended questions which asked the educator to reflect on activity provision for the child as well as what physical activity means for themselves. Qualitative data will be explored in a future publication since it does not contribute to the structural validation of this tool.

Knowledge items are scored correct (1) or incorrect (0) and summed to derive the total score out of 7 with a higher score indicating stronger knowledge of physical literacy concepts.

**Table 4. Correlations among factors in the five-factor model.**

|  | Factor 1 | Factor 2 | Factor 3 | Factor 4 |
|---|---|---|---|---|
| Factor 2 | 0.39 |  |  |  |
| Factor 3 | 0.53 | 0.55 |  |  |
| Factor 4 | 0.38 | 0.41 | 0.38 |  |
| Factor 5 | 0.28 | 0.30 | 0.37 | 0.25 |

**Table 5. Cronbach's alpha table of various aspects of the PLKASB-ECE.**

| PLKASB-ECE | Cronbach's Alpha | Ordinal Omega |
|---|---|---|
| Perception of Knowledge (2 items) | 0.819 | 0.819 |
| Attitude (3 items) | 0.711 | 0.723 |
| Self-Efficacy (3 items) | 0.783 | 0.783 |
| Environment Behaviors (3 items) | 0.774 | 0.779 |
| Personal Behaviors (3 items) | 0.701 | 0.735 |
| All Perception of Knowledge, Self-Efficacy, Attitude and Behavior Items (14 items) | 0.855 | 0.841 |
| All Perception of Knowledge, Self-Efficacy, Attitude and Behavior Items (14 items) with Knowledge Sum Score (1 item) | 0.851 | 0.840 |

PLKASB–ECE, physical literacy knowledge, attitudes, self-efficacy, and behaviors of early childhood educator

All perception, self-efficacy, attitudinal, and behavior items employ a 7-point Likert scale anchored by "strongly disagree" scored as a one at one end of the scale and "strongly agree" scored as a seven at the other end of the scale with one exception: item #20 is scored 0 to 7. A composite score can be calculated by adding the knowledge items (n = 7) to all of the items from the five subdomains (n = 14), plus one additional value to account for item #20: ((7 x 1) + (14 x 7) + 1) = 106. Once the composite score is calculated, it can be converted to a percent score. For example, if an educator scored 85/106, their composite score would be 80.2%.

**Table 6. Five-factor model definitions and questionnaire items.**

| Construct | Definition | Items |
|---|---|---|
| Perception of Knowledge | Educator perception of their physical literacy knowledge. | understand what physical competence means |
| | | know what fundamental movement skills are |
| Attitude | Educator attitudes regarding physical literacy and positive health outcomes. | working on physical literacy sets a child up for being a physically active adult |
| | | support children to be physically active in our center |
| | | building my physical literacy skills can support my personal health |
| Self-Efficacy | Educator self-efficacy towards developing physical literacy in the ECE environment. | confident including movement skills in my daily practice |
| | | confident identifying when a child is struggling with a movement skill |
| | | confident giving children strategies to improve their movement skills |
| Environment Behaviors | Educator physical literacy practices in the ECE environment. | plan 2 or more INDOOR physical activity experiences each day |
| | | plan 2 or more OUTDOOR physical activity experiences each day |
| | | ask the children to tell me what they are learning when they engage in PA |
| Personal Behaviors | Personal physical literacy practices. | do 30 minutes of heart pumping physical activity ___ days per week |
| | | plan to find time to include physical activity daily for myself |
| | | enjoy being physically active. |

Interpretation of the percent score would indicate higher levels of understanding, knowledge, attitudes and behaviors of the physical literacy construct, generally. Additionally, there may be utility in using the individual constructs for research or intervention purposes.

## Discussion

This study addressed the development and validation of a comprehensive instrument to measure educator knowledge, attitudes, self-efficacy, and behavior capabilities to foster physical literacy in young children within the ECEC environment. Content validity was conducted using an iterative process with experts with the final questions achieving good I-CVI after three rounds of discussion. Factor analysis revealed five distinct subscales beyond the knowledge-based items in the overall PLKASB-ECE tool: 1) perception of own knowledge regarding physical literacy; 2) attitude regarding physical literacy and positive health outcomes; 3) self-efficacy of skills regarding physical literacy teaching; 4) environment behaviors revealing educator practices in the ECE environment; and 5) personal behaviors revealing educator's own physical literacy practices. These factors loaded independently and were weakly to moderately correlated, suggesting they represent related but distinct constructs that can be interpreted separately by domain. The overall value from the PLKASB-ECE may also have utility as a single value. This is further supported by good internal consistency of the 14-items (Cronbach's alpha = 0.86) indicating a strong underlying construct measure for the holistic nature of physical literacy. The tool readability was finalized at a Grade 8.7 (F-K score) and most participants completed the 25 items in 8.15 minutes (IQR 5.56–14.16) indicating reasonable usability.

Results from our structured literature search (Phase 1 –item generation) for a holistic assessment tool to evaluate educator physical literacy capabilities within ECECs revealed that no single tool exists to measure multiple physical literacy domains of educators within their teaching context. This gap in the literature is supported by a recent study investigating a physical literacy professional development intervention for early childhood educators, where authors highlighted the limitation of not having a single instrument to evaluate multiple domains of physical literacy in this population [36]. To our knowledge, no valid and reliable comprehensive instrument is currently available to assess educators' knowledge, attitudes, self-efficacy, and behaviors related to fostering children's physical literacy development. Thus, this tool addresses a gap identified in the literature and provides a comprehensive assessment for interventions aimed at enhancing educators' capacity to support physical literacy in children within ECECs.

Measurement of early childhood physical literacy presents unique challenges due to the varied developmental stages within this age group, where children grow and learn at different rates [78]. To address these measurement challenges, we propose assessing educators' personal and professional perspectives of physical literacy as a proxy for the environment in which young children engage in their physical literacy journey. This approach aligns with the conceptual model of an effective early childhood physical literacy pedagogue. This model suggests that an educators' perceived physical literacy capabilities, knowledge, and practices to promote physical literacy are measurable indicators of their effectiveness [34]. We support the assertion that educators have considerable impact on children's physical literacy development [79–82]. This type of influence on the early behaviors in the physical competence domain of physical literacy has been demonstrated by Barnett and colleagues [83,84]. However, only future longitudinal research, following children over time and examining the influences they experienced in their early years, will answer questions about long-term impact.

The four elements of physical literacy within Canada's Consensus framework include the cognitive, affective, behavioral, and physical domains [17]. The PLKASB-ECE factor structure

fits within three of the four framework elements. The cognitive domain encompassing the knowledge, understanding, and ability to conceptualize and express movement qualities and how they relate to health [17]. The PLKASB-ECE subscales of Knowledge and Perception of Knowledge address this domain. The affective domain refers to the motivation and confidence to incorporate movement into daily life [17]. The PLKASB-ECE subscales of Attitude and Self-Efficacy address belief that physical activity supports health as well as confidence in being able to include it within daily practice. As such these subscales fit within the affective domain. The behavioral domain addresses the engagement in physical activities on a daily basis [17]. The PLKASB-ECE subscales of Environmental Behavior and Personal Behavior speak to the educator actions related to taking responsibility for physical literacy experiences with their classroom setting as well as including physical activity in their personal daily practice which align with the behavioural domain. Finally, the physical domain covers the skills and competence required to participate in various activities [17]. We identified existing tools in the literature targeting fundamental skill development for preschool (3–5 years) children that relate to this physical competence [54–57] as this aspect of physical literacy was not included in the PLKASB-ECE.

Only a handful of assessment tools have been developed to assess physical literacy for the early childhood educator population. A systematic review of tools measuring self-efficacy in the ECEC context identified six tools addressing physical literacy content; however, most focused primarily on fundamental movement skills and only two included measurement of confidence [33]. Buckler and Bredin [35] created a survey to examine educators' physical literacy knowledge, competence, and confidence to better understand the ECEC physical literacy environment, but they noted that the survey did not undergo a validation process. In a second study, Buckler and colleagues [80] assessed educators' physical competence, understanding, motivation, confidence, intentions, and behaviors using five different tools. Similarly, Simpson and colleagues [36] employed a combination of open-ended question responses scored by research team members, and four separate questionnaires to assess behavior, perceived value, confidence in teaching and planning physical literacy. In our study we have taken a similar holistic assessment approach to include multiple domains within a single comprehensive tool and we provide evidence of its validity and reliability indices within this population.

Bruijn and colleagues conducted a validation study to develop a tool to measure educators' behavioral intention and perceived behavioral control to promote activity and outdoor play within childcare environments [79]. In addition, they conducted a validation study to develop a tool to measure educators' self-efficacy to promote activity and outdoor play within childcare environments [85]. In our study, we measure similar but slightly different constructs. Our focus encompasses personal knowledge, attitude, self-efficacy and workplace behaviors with a focus on physical literacy. Again, ideally, this same type of longitudinal tracking of a more holistic perspective of physical literacy will help to measure the success of those early year interventions. A tool that measures those physical literacy foundations with early childhood educators is an important first step in understanding this piece of the journey.

## Strengths and limitations

To our knowledge this is the first tool to measure the physical literacy knowledge, attitude, self-efficacy and practice composite construct for early childhood educators in the early learning and child care environment. Validation of this tool included a large geographical region and alignment with provincial population distributions. In addition, the recruited centers were representative of child care centers across a large geographical region and alignment with provincial population distribution. The proportion of recruited centers (large urban, medium,

and small population centers) were relatively similar to overall population distribution across the province [45]. Some limitations should be considered when interpreting the findings of this study. First, the study sample population was limited to the province of Alberta, Canada which limits the generalizability of the findings. Second, most participants were women. However, this proportion is similar to the ECEC workforce in Canada so this sampling is reflective of the intended population [86]. Third, respondent bias is possible given that data were self-reported. Fourth, the management of including knowledge into a factor model is challenging since the items are graded as correct or incorrect, but the collection of the seven items is clearly a univariate measure. Most EFAs have alternative methods to manage or include the knowledge items relative to the other constructs [87]. Our plan was to run the EFA with a single knowledge value knowing it likely would not factor into the model as a separate subdomain. However, we have included knowledge in the composite or percent score for a more holistic representative measure of physical literacy. Lastly, the factor structure of the PLKASB-ECE has not been validated with a new independent sample using confirmatory factor analysis. The sample size in the current study was not sufficient to partition for dual exploratory and confirmatory factor analysis. The next step in the iterative process of scale development for PLKAS-B-ECE is to conduct confirmatory factor analysis with an independent sample to substantiate the factor structure indicated in this study.

## Conclusion

This study described the development and validation towards the psychometric assessment of the PLKASB-ECE questionnaire as tool to measure educators' knowledge, attitude, self-efficacy, and both personal and ECEC related physical literacy concepts. High internal consistency and good factorial validity indicate the PLKASB-ECE can provide an overall assessment of the physical literacy construct with an early childhood educator context as well as the five subscales for evaluating the physical literacy setting within an ECEC environment.

## Supporting information

**S1 Appendix. Content validity of items on the PLKASB-ECE.**
(PDF)

**S2 Appendix. PLKASB-ECE questionnaire items.**
(PDF)

## Acknowledgments

We thank the early childhood educators who took part in this study. Their dedication and passion to support young children inspires us.

## Author Contributions

**Conceptualization:** Lynne M. Z. Lafave, Nadine Van Wyk, Alexis D. Webster, Mark R. Lafave.

**Data curation:** Lynne M. Z. Lafave, Nadine Van Wyk, Alexis D. Webster, Joyce Hayek.

**Formal analysis:** Lynne M. Z. Lafave, Nadine Van Wyk, Alexis D. Webster, Joyce Hayek, Mark R. Lafave.

**Funding acquisition:** Lynne M. Z. Lafave.

**Investigation:** Lynne M. Z. Lafave.

**Methodology:** Lynne M. Z. Lafave, Nadine Van Wyk, Alexis D. Webster, Mark R. Lafave.

**Project administration:** Lynne M. Z. Lafave, Nadine Van Wyk, Alexis D. Webster, Joyce Hayek.

**Resources:** Lynne M. Z. Lafave.

**Supervision:** Lynne M. Z. Lafave, Joyce Hayek.

**Validation:** Lynne M. Z. Lafave.

**Writing – original draft:** Lynne M. Z. Lafave, Alexis D. Webster, Mark R. Lafave.

**Writing – review & editing:** Lynne M. Z. Lafave, Nadine Van Wyk, Alexis D. Webster, Joyce Hayek, Mark R. Lafave.

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
