## [Decision Letter · Decision Letter 0]

7 May 2024

PONE-D-24-06642Developing a Physical Literacy Knowledge, Attitudes, Self-Efficacy, and Behaviors Questionnaire for Early Childhood Educators (PLKASB-ECE): An Exploratory Factor AnalysisPLOS ONE

Dear Dr. Lafave,

Thank you for submitting your manuscript to PLOS ONE. After careful consideration, we feel that it has merit but does not fully meet PLOS ONE’s publication criteria as it currently stands. Therefore, we invite you to submit a revised version of the manuscript that addresses the points raised during the review process.

**thank you for permitting us to review your manuscript.**

Please include the following items when submitting your revised manuscript:A rebuttal letter that responds to each point raised by the academic editor and reviewer(s). You should upload this letter as a separate file labeled 'Response to Reviewers'.A marked-up copy of your manuscript that highlights changes made to the original version. You should upload this as a separate file labeled 'Revised Manuscript with Track Changes'.An unmarked version of your revised paper without tracked changes. You should upload this as a separate file labeled 'Manuscript'.

We look forward to receiving your revised manuscript.

Kind regards,

Mary Diane Clark, PhD

Academic Editor

PLOS ONE

Journal Requirements:

2. In the online submission form, you indicated that The datasets generated during and/or analyzed during the current study are not publicly available due to participants’ privacy under the REB approval. Requests to access the datasets should be directed to llafave@mtroyal.ca.

3. We note that this data set consists of interview transcripts. Can you please confirm that all participants gave consent for interview transcript to be published?

If they DID provide consent for these transcripts to be published, please also confirm that the transcripts do not contain any potentially identifying information (or let us know if the participants consented to having their personal details published and made publicly available). We consider the following details to be identifying information:

- Names, nicknames, and initials

- Age more specific than round numbers

- GPS coordinates, physical addresses, IP addresses, email addresses

- Information in small sample sizes (e.g. 40 students from X class in X year at X university)

- Specific dates (e.g. visit dates, interview dates)

- ID numbers

Or, if the participants DID NOT provide consent for these transcripts to be published:

- Provide a de-identified version of the data or excerpts of interview responses

- Provide information regarding how these transcripts can be accessed by researchers who meet the criteria for access to confidential data, including:

a) the grounds for restriction

b) the name of the ethics committee, Institutional Review Board, or third-party organization that is imposing sharing restrictions on the data

c) a non-author, institutional point of contact that is able to field data access queries, in the interest of maintaining long-term data accessibility.

d) Any relevant data set names, URLs, DOIs, etc. that an independent researcher would need in order to request your minimal data set.

For further information on sharing data that contains sensitive participant information, please see: https://journals.plos.org/plosone/s/data-availability#loc-human-research-participant-data-and-other-sensitive-data

If there are ethical, legal, or third-party restrictions upon your dataset, you must provide all of the following details (https://journals.plos.org/plosone/s/data-availability#loc-acceptable-data-access-restrictions):

a. A complete description of the dataset

b. The nature of the restrictions upon the data (ethical, legal, or owned by a third party) and the reasoning behind them

c. The full name of the body imposing the restrictions upon your dataset (ethics committee, institution, data access committee, etc)

d. If the data are owned by a third party, confirmation of whether the authors received any special privileges in accessing the data that other researchers would not have

e. Direct, non-author contact information (preferably email) for the body imposing the restrictions upon the data, to which data access requests can be sent

Additional Editor Comments:

Please see the reviewers suggestions to improve the paper. There ae many issues with the statistics so please work with someone to satisfy these concerns of the reviewers

This revise and resubmit option does not include a promise that the revision will be accepted, but I am willing to give you a chance. I think that both reviewers give you excellent feedback about revising your paper and mostly focusing on the statistics.

Reviewers' comments:

Reviewer's Responses to Questions

**Comments to the Author**

1. Is the manuscript technically sound, and do the data support the conclusions?

Reviewer #1: Partly

Reviewer #2: No

2. Has the statistical analysis been performed appropriately and rigorously? 

Reviewer #1: No

Reviewer #2: No

3. Have the authors made all data underlying the findings in their manuscript fully available?

Reviewer #1: Yes

Reviewer #2: No

4. Is the manuscript presented in an intelligible fashion and written in standard English?

Reviewer #1: No

Reviewer #2: Yes

5. Review Comments to the Author

**Reviewer #1**: ID: PONE-D-24-06642

Title: Developing a Physical Literacy Knowledge, Attitudes, Self-Efficacy, and Behaviors Questionnaire for Early Childhood Educators (PLKASB-ECE): An Exploratory Factor Analysis

Thank you for providing a chance to review this manuscript.

Detailed information:

Title

Overall: The study includes both development and validation components, and the process of “validation of the questionnaire” should be emphasized in the title, in addition to the relevant description of the questionnaire development.

Abstract

Overall: Overall, as an Abstract, there are issues of length and redundancy, and the authors are advised to further minimize unnecessary descriptions.

Line 24-25, Page 2: “subsequently underwent several rounds of expert content validation……”, write in detail the number of rounds for which expert content validation is performed; vague terms are not recommended.

Line 28-29, Page 2: The time period for validation sample collection was? Please add any relevant information.

Overall: “PLKASB-ECE”, is this an abbreviated form of the scale? Didn't find an explanation of the full name in the Abstract.

Keywords: Quantitatively, there are too many keywords set, and it is recommended that some that are not relevant to the focus of the study be removed.

Introduction

Line 38, Page 2: How does the author define “physical activity” before describing the phenomenon in question?

Line 47-56, Page 3: Are the disadvantages of physical inactivity or the benefits of physical activity specific to the group of childhood aged 0-5 years? It is recommended that the authors add relevant statements, which will help to reflect the need to study the population.

Line 62-64, Page 3: Elaborating on the mechanisms by which these four interrelated domains (affective, physical, cognitive, behavioral) contribute to the holistic development of physical literacy facilitates a clearer presentation and is recommended to be added by the authors.

Line 64-66, Page 3: More presentations of relevant mechanisms, or more detailed descriptions are necessary.

Line 71-73, Page 4: “These settings provide a fertile ground for targeting strategies for nurturing physical literacy in the early years”, how do these environmental factors affect the quality of education? Related content suggestions are presented.

Line 76, Page 4: What does “ECEC” mean?

Line 73-79, Page 4: Is the professional preparation of early childhood educators only about education? The relevant descriptions are limited.

Line 92-93, Page 5: “In practice, very little time is available during the workday for early childhood educators to pursue this task”, is there a basis for supporting this view in detail?

Line 95-100, Page 5: The representation of relevant measures is inadequate, and the importance of a brief and valid assessment tool for physical literacy for early childhood educators has not been adequately captured.

Line 99, Page 5: “Self-efficacy” is not an unusual term and it is suggested that a definition be added.

Overall: There is too little about how your study compares to other similar studies to highlight the innovative nature of your study, and additions are recommended.

Methods

Line 105, Page 5: The setting of subheading case should be standardized throughout the text, please note and revise the relevant sections.

Line 110-112, Page 5 ＆ Line 113-124, Page 6: Does the search keywords include only “physical literacy”, without considering other similar terms?

Line 117-121, Page 6: An introduction to ECEC content would be more appropriately placed in a separate paragraph of the introduction.

Face validity, Page 7: Please specifically describe the assessment process, criteria, and the handling of the results.

Line 144-146, Page 7: Please add the treatment when evaluating readability, when F-K and FRE show consistent or discrepant results.

Line 156-158, Page 7 ＆ Line 159-160, Page 8: The number of ECEC centers selected and their proportion in large urban population centers, medium population population centers and low population centers should likewise be carefully accounted for.

Line 160-163, Page 8: “In the first year of data collection, center directors were invited to participate through phone correspondence, provided with general information on the study and allocated to either intervention or control group based on center directors’ assessment of both available time and capacity.” Please tell us what the criteria were for the center director to be assigned to the intervention or control group.

Line 160-171, Page 8: The “Participants” section is overly oriented towards the screening of ECEC centers. Are there clear inclusion and exclusion criteria for participants?

Line 169-171, Page 8: “Each participating center was requested to identify three staff members to be included in the study, with a preference to include two educators and the center director or manager.” How were these two educators chosen? Also how are they elected if there is more than one center director or manager in the participating centers?

Line 184-186, Page 9: “In exploratory factor analysis a recommended minimum subject-to-item (N:p) ratio ranges from 5:1 to 10:1” How do you determine that this is the optimal ratio? Please add supporting literature.

Page 8: Expressions related to ethical norms such as informed consent of participants are needed to be added.

Results

Participants, Page 12: “78.5% of these centers were from large urban population cities”, does this; distribution of centers lead to bias due to unevenness?

Line 253-254, Page 12: “After recording and editing the readability of the final items ranged from 5.2 to 9.5.” I don't quite understand how this process is accomplished, please explain in detail.

Table 1, Page 13: Comparing the unbalanced sex ratio, the sample is very skewed towards males, is this similar to the overall distribution of genders in the group?

Discussion

Line 365-366, Page 22: “This tool addresses a gap that has been identified for the early childhood (i.e. 0-5) community while also trying to address a holistic measurement of physical literacy from both a qualitative and quantitative perspective.” All of your studies were of young children aged 2-5 years, and it seems inappropriate to say 0-5 years here.

I have thoroughly reviewed the entire manuscript, and I must commend the authors for conducting an exhaustive study. The authors have undertaken a substantial amount of work, which is undeniable. Nevertheless, a number of issues remain to be noted, modified, added or deleted. I hope the authors will address the comments comprehensively in the final version, enhancing the depth and persuasiveness of the research.

Thank you and my best

Your reviewer

**Reviewer #2**: Thank you for the opportunity to review the manuscript PONE-D-24-06642.e.

The study holds practical significance, but the analysis results are dubious for the following reasons:

The exploratory factor analysis (EFA) is incomplete. As per Table 3, the item "As a child, I was physically active outside of school time" lacks a satisfactory factor loading on any factor, indicating a weak correlation with other items. This item should be removed, followed by a revised EFA to assess if the remaining items still support a 5-factor model.

Additionally, the factor structure of the Physical Literacy Knowledge, Attitudes, Self-Efficacy, and Behaviors Questionnaire has not been validated with a new sample using confirmatory factor analysis. Consequently, it's unclear if the 5-factor model is optimal, despite initial EFA findings suggesting its promise.

The unresolved factor structure issue undermines the robustness of the study's conclusions and limits its applicability.

6. PLOS authors have the option to publish the peer review history of their article (what does this mean?). If published, this will include your full peer review and any attached files.

Reviewer #1: No

Reviewer #2: No

---

## [Author Response · Author response to Decision Letter 0]

27 May 2024

PONE-D-24-06642

Development and validation of a Physical Literacy Knowledge, Attitudes, Self-Efficacy, and Behaviors Questionnaire for Early Childhood Educators (PLKASB-ECE): An Exploratory Factor Analysis

PLOS ONE

Author Response: Thank you for offering us an opportunity to revise our paper “Validity and reliability of a physical literacy knowledge, attitudes, self-efficacy and behaviors questionnaire for early childhood educators (PLKASB-ECE): An exploratory factor analysis” (PONE-D-24-06642). We have included a response to reviewers in which we address each comment the reviewers made. The remarks of the reviewers are numbered and depicted first, followed by our responses to these comments. Corresponding changes are highlighted in the manuscript text.

Reviewer #1: ID: PONE-D-24-06642

Overall 

Reviewer #1 Comment #1: The study includes both development and validation components, and the process of “validation of the questionnaire” should be emphasized in the title, in addition to the relevant description of the questionnaire development.

Author Response (R1-C1): Thank you for this suggestion. It is an important element of our study and we have edited the title as suggested. The title now reads “Validity and reliability of a physical literacy knowledge, attitudes, self-efficacy and behaviors questionnaire for early childhood educators (PLKASB-ECE): An exploratory factor analysis” 

Abstract

Reviewer #1 Comment #2: Overall, as an Abstract, there are issues of length and redundancy, and the authors are advised to further minimize unnecessary descriptions.

Author Response (R1-C2): We have reviewed the abstract with this suggestion in mind. We revised the abstract to remove redundancies and clarified the details of the items in the revised EFA. These changes are reflected in the track changes manuscript. 

Reviewer #1 Comment #3: Line 24-25, Page 2: “subsequently underwent several rounds of expert content validation……”, write in detail the number of rounds for which expert content validation is performed; vague terms are not recommended.

Author Response (R1-C3): We agree. We have revised the sentence with specific details and Line 31 now reads “subsequently underwent three rounds of expert content validation……”

Reviewer #1 Comment #4: Line 28-29, Page 2: The time period for validation sample collection was? Please add any relevant information.

Author Response (R1-C4): Thank you for catching this detail. We have aligned the time period for sample collection with the methods section. Line 35-36 now reads: “The PLKASB-ECE tool was administered to 470 educators in Alberta, Canada between 2019 to 2022.”

Reviewer #1 Comment #5: Overall: “PLKASB-ECE”, is this an abbreviated form of the scale? Didn't find an explanation of the full name in the Abstract. 

Author Response (R1-C5): Thank you for identifying this. We have changed the abstract ordering of words and addition of the acronym to better align with the name of the tool. Line 26 now reads: “The purpose of this study was to develop a holistic digital tool to measure physical literacy knowledge, attitudes, self-efficacy, and behaviors of early childhood educator (PLKASB-ECE) for both their professional teaching context as well as the educator’s own personal context.”

Reviewer #1 Comment #6: Keywords: Quantitatively, there are too many keywords set, and it is recommended that some that are not relevant to the focus of the study be removed.

Author Response (R1-C6): We have removed keyword sets and there are now 5 keyword sets

Introduction

Reviewer #1 Comment #7: Line 38, Page 2: How does the author define “physical activity” before describing the phenomenon in question?

Author Response (R1-C7): Thank you for this suggestion. We have included a definition at the beginning of the introduction to establish context. Line 44 now reads: “Physical activity, bodily movement resulting from skeletal muscle activation leading to increased energy expenditure, has been identified as a protective factor for human health” 

Reviewer #1 Comment #8: Line 47-56, Page 3: Are the disadvantages of physical inactivity or the benefits of physical activity specific to the group of childhood aged 0-5 years? It is recommended that the authors add relevant statements, which will help to reflect the need to study the population.

Author Response (R1-C8): We agree. We have reorganized the introduction to bring the paragraph of the health benefits of physical activity into the first paragraph and included the disadvantages of physical inactivity related to NCDs in this first paragraph. Line 51-56 now provide both the benefits of physical activity and disadvantages of physical inactivity. 

Reviewer #1 Comment #9: Line 62-64, Page 3: Elaborating on the mechanisms by which these four interrelated domains (affective, physical, cognitive, behavioral) contribute to the holistic development of physical literacy facilitates a clearer presentation and is recommended to be added by the authors.

Author Response (R1-C9): Thank you for this suggestion. We have added the following to better describe this important concept: Line 93 “This can be conceptualized in a framework of four interrelated domains (affective, physical, cognitive, behavioural) that contribute to the holistic development of physical literacy that consider the social processes associated with lifelong learning and prioritization in lifelong physical activity engagement (Cairney et al., 2019; Cornish et al., 2020). It has been proposed that a strong physical literacy foundation will improve an individual’s overall health and wellness through the building of an individual’s positive relationship with daily movement (Cornish et al., 2020; Edwards et al., 2017; Shearer et al., 2021; Wainwright et al., 2020). 

Reviewer #1 Comment #10: Line 64-66, Page 3: More presentations of relevant mechanisms, or more detailed descriptions are necessary.

Author Response (R1-C10): We have improved the connections and details as described above (R1-C9).

Reviewer #1 Comment #11: Line 71-73, Page 4: “These settings provide a fertile ground for targeting strategies for nurturing physical literacy in the early years”, how do these environmental factors affect the quality of education? Related content suggestions are presented.

Author Response (R1-C11): Thank you for this comment. With the restricting of the introduction to improve flow we connect physical activity to improved health outcomes that can be taught through a physical literacy lens that is provided in high-quality child care environments. Line 96 now reads: “High quality child care has long-term effects on a child’s cognitive, academic, and language development tracking into adolescence (Vandell et al., 2010). Early childhood educators’ health knowledge and practices contribute to high quality care and are vital in creating a health-supportive environment, influencing a child's lifelong relationship with physical activity (Lugossy et al., 2022; World Health Organization, 2016).” 

Reviewer #1 Comment #12a: Line 76, Page 4: What does “ECEC” mean? Line 73-79, Page 4: Is the professional preparation of early childhood educators only about education? The relevant descriptions are limited.

Author Response (R1-C12): ECEC is defined earlier in the manuscript, but only in the abstract. We have added the full written component to the Introduction: Line 79-80 now reads “early childhood education and care (ECEC) settings”

Reviewer #1 Comment #12b: Line 73-79, Page 4: Is the professional preparation of early childhood educators only about education? The relevant descriptions are limited.

Author Response (R1-C12b): Thank you for pointing this out. The descriptions of the education preparation of educators require clarity. We have revised the description for clarity - Line 101-103 “… some educators have completed an early childhood education certificate after completing high school whereas others report having college degrees with early childhood education diplomas.” This description aligns with demographic descriptions presented in Table 1.

Reviewer #1 Comment #13: Line 92-93, Page 5: “In practice, very little time is available during the workday for early childhood educators to pursue this task”, is there a basis for supporting this view in detail?

Author Response (R1-C13): Thank you for the clarification request. We see this in our work regularly and it is further outlined in the literature. We have added three references to support educators’ time limitation barrier to professional development. Line 127-128 now reads “In practice, very little time is available during the workday for early childhood educators to pursue this task (McKinlay et al., 2018; Brussoni et al., 2021; Pölzl-Stefanec, 2021).” 

Reviewer #1 Comment #14: Line 95-100, Page 5: The representation of relevant measures is inadequate, and the importance of a brief and valid assessment tool for physical literacy for early childhood educators has not been adequately captured.

Author Response (R1-C14): We have included an additional systematic review looking for assessment tools in this area that concludes there is a paucity of tools available for assessing early childhood educators as a proxy for the child care setting in promoting physical activity in the young child. We add this article to provide the rationale for a brief and valid assessment tool for physical literacy for early childhood educators. Line 132-135 now reads: “In another systematic review it was noted that there was limited research for assessment tools reporting validity and reliability measures available to measure early childhood educators’ self-efficacy in supporting young children’s physical activity and reduction in sedentary behaviour (Szpunar et al., 2021).”

Reviewer #1 Comment #15: Line 99, Page 5: “Self-efficacy” is not an unusual term and it is suggested that a definition be added.

Author Response (R1-C15): We have included a definition to provide more clarity on the use of ‘self-efficacy’ within a social cognitive theory framework by adding Line 115-121 “Social cognitive factors have been used to understand why people engage in certain behaviors and are believed to predict intentional behavior (Ajzen, 1991; Bandura, 2004; De Vries, 2017; Janz & Becker, 1984). Social cognitions include constructs such as attitude and efficacy beliefs. Attitude is defined as the perceived favorable or unfavorable evaluation of a certain behavior (Ajzen & Fishbein, 2002), while self-efficacy refers to the belief in one’s ability to perform a particular behavior (Bandura, 1977). In the context of physical literacy and early childhood education, research suggests that efficacy beliefs of educators can influence their ability and intention to facilitate movement for children (Szpunar et al., 2021; Copeland et al., 2012).”

Reviewer #1 Comment #16: Overall: There is too little about how your study compares to other similar studies to highlight the innovative nature of your study, and additions are recommended.

Author Response (R1-C16): Thank you for suggesting we highlight the innovative nature of our study. We have provided context with the changes described in Comment #14. This is also explored in great detail in Phase 1 of the results section as it is part of the needs assessment portion of the methods. We also address this in the discussion section when contextualizing the PLKASB-ECE in the context of the literature that exists currently.

Methods

Reviewer #1 Comment #17: Line 105, Page 5: The setting of subheading case should be standardized throughout the text, please note and revise the relevant sections.

Author Response (R1-C17): Thank you for identifying this. We have revised the headings to be standardized throughout the manuscript.

Reviewer #1 Comment #18: Line 110-112, Page 5 ＆ Line 113-124, Page 6: Does the search keywords include only “physical literacy”, without considering other similar terms?

Author Response (R1-C18): The search term used was “physical literacy” along with truncated versions such as “physical lit*” to focus the search on physical literacy concepts and avoid confusion with ‘play’ or ‘physical activity’. This structure is similar to search strategies used in Physical Literacy Systematic Reviews conducted by Edwards et al., 2018; Jean de Dieu & Zhou, 2021; Carl et al., 2022. 

• Jean de Dieu, H., & Zhou, K. (2021). Physical literacy assessment tools: a systematic literature review for why, what, who, and how. International Journal of Environmental Research and Public Health, 18(15), 7954.

• Carl, J., Barratt, J., Toepfer, C., Cairney, J., & Pfeifer, K. (2022). How are physical literacy interventions conceptualized?–a systematic review on intervention design and content. Psychology of Sport and Exercise, 58, 102091.

• Edwards, L. C., Bryant, A. S., Keegan, R. J., Morgan, K., Cooper, S. M., & Jones, A. M. (2018). ‘Measuring’ physical literacy and related constructs: A systematic review of empirical findings. Sports Medicine, 48, 659-682.

Reviewer #1 Comment #19: Line 117-121, Page 6: An introduction to ECEC content would be more appropriately placed in a separate paragraph of the introduction.

Author Response (R1-C19): Thank you for this observation. We have moved the ECEC content from page 6 to the Introduction. See Line 110 - 121

Reviewer #1 Comment #20: Face validity, Page 7: Please specifically describe the assessment process, criteria, and the handling of the results.

Author Response (R1-C20): We have changed the sentence for clarity. Line 171-175 now reads “To address face validity of the scale, two in-service ECE educators, two pre-service ECE students, and two ECE university faculty members were identified for their expertise to provide a stakeholder perspective. The survey questions were provided to individuals to read and the research team took notes of verbal feedback regarding item clarity, comprehension, and suitability to the community.”

Reviewer #1 Comment #21: Line 144-146, Page 7: Please add the treatment when evaluating readability, when F-K and FRE show consistent or discrepant results.

Author Response (R1-C21): The readability score report from Microsoft Word software is provided in a single report. The equations to compute readability are incorporated into versions of Microsoft Word and Zhou et al., (2017) suggest using Microsoft Word when Grade Level is needed as it makes the fewest error in linguistic elements. We have added the following for clarity Line 179-180: “The F-K score was the primary outcome in evaluating readability.” [Zhou, S., Jeong, H., & Green, P. A. (2017). How consistent are the best-known readability equations in estimating the readability of design standards? IEEE Transactions on Professional Communication, 60(1), 97-111.]

Reviewer #1 Comment #22: Line 156-158, Page 7 ＆ Line 159-160, Page 8: The number of ECEC centers selected and their proportion in large urban population centers, medium population population centers and low population centers should likewise be carefully accounted for.

Author Response (R1-C22): Thank you for this comment. We intended to communicate our effort to recruit child care centers throughout the province and with representation in various size communities. We have added a sentence to reflect the effort of this recruitment strategy. Line 195--198 now reads: “Large urban population centers represent approximately 70% of the provinces population with 16% and 13% falling in the medium and small population centers, respectively (Government of Canada, 2017). We looked to balance the proportion of center recruitment relative to the overall provincial population.”

Reviewer #1 Comment #23: Line 160-163, Page 8: “In the first year of data collection, center directors were invited to participate through phone correspondence, provided with general information on the study and allocated to either intervention or control group based on center directors’ assessment of both available time and capacity.” Please tell us what the criteria were for the center director to be assigned to the intervention or control group.

Author Response (R1-C23): All data for this study is derived 

---

## [Decision Letter · Decision Letter 1]

9 Sep 2024

PONE-D-24-06642R1Validity and reliability of a physical literacy knowledge, attitudes, self-efficacy and behaviors questionnaire for early childhood educators (PLKASB-ECE): An exploratory factor analysisPLOS ONE

Dear Dr. Lafave,

Thank you for submitting your manuscript to PLOS ONE. After careful consideration, we feel that it has merit but does not fully meet PLOS ONE’s publication criteria as it currently stands. Therefore, we invite you to submit a revised version of the manuscript that addresses the points raised during the review process.

Please include the following items when submitting your revised manuscript:A rebuttal letter that responds to each point raised by the academic editor and reviewer(s). You should upload this letter as a separate file labeled 'Response to Reviewers'.A marked-up copy of your manuscript that highlights changes made to the original version. You should upload this as a separate file labeled 'Revised Manuscript with Track Changes'.An unmarked version of your revised paper without tracked changes. You should upload this as a separate file labeled 'Manuscript'.If applicable, we recommend that you deposit your laboratory protocols in protocols.io to enhance the reproducibility of your results. Protocols.io assigns your protocol its own identifier (DOI) so that it can be cited independently in the future. For instructions see: https://journals.plos.org/plosone/s/submission-guidelines#loc-laboratory-protocols. Additionally, PLOS ONE offers an option for publishing peer-reviewed Lab Protocol articles, which describe protocols hosted on protocols.io. Read more information on sharing protocols at https://plos.org/protocols?utm_medium=editorial-email&utm_source=authorletters&utm_campaign=protocols.

We look forward to receiving your revised manuscript.

Kind regards,

Mary Diane Clark, PhD

Academic Editor

PLOS ONE

Journal Requirements:

Additional Editor Comments:

Thank you for clarifying the comments form the earlier reviewers. When I sent it out again, there was one of the original reviewers and a new reviewer. They are both asking for some modifications. Your clarification of the results seems to have been really effective as the new reviewer did not comment on that section of the manuscript. Regardless both have a some issues that will make it a more usable publication.

I have one request. Can you make your headings the same font and font size as the other parts of the paper.

We look forward to this next revision.

Reviewers' comments:

Reviewer's Responses to Questions

**Comments to the Author**

1. If the authors have adequately addressed your comments raised in a previous round of review and you feel that this manuscript is now acceptable for publication, you may indicate that here to bypass the “Comments to the Author” section, enter your conflict of interest statement in the “Confidential to Editor” section, and submit your "Accept" recommendation.

Reviewer #1: (No Response)

Reviewer #3: (No Response)

2. Is the manuscript technically sound, and do the data support the conclusions?

Reviewer #1: Yes

Reviewer #3: Yes

3. Has the statistical analysis been performed appropriately and rigorously? 

Reviewer #1: Yes

Reviewer #3: I Don't Know

4. Have the authors made all data underlying the findings in their manuscript fully available?

Reviewer #1: Yes

Reviewer #3: Yes

5. Is the manuscript presented in an intelligible fashion and written in standard English?

Reviewer #1: Yes

Reviewer #3: Yes

6. Review Comments to the Author

Reviewer #1: ID: PONE-D-24-06642R1

Title: Validity and reliability of a physical literacy knowledge, attitudes, self-efficacy and behaviors questionnaire for early childhood educators (PLKASB-ECE): An exploratory factor analysis

Thank you for providing a chance to review this manuscript.

Recommendation: Minor revision.

The author has made careful revisions and the quality of the article has been greatly improved, congratulations! However, some detailed issues still need to be improved. I also have the following minor issues to express my doubts:

Detailed information:

Abstract

Overall: Subheadings can be used to separate the content of the abstract, including “Background”, “Objective”, “Methods”, “Results”, “Conclusion”, etc.

Introduction

Line 66-74, Page 4: Perhaps it would make more sense to present the situation of the group of early childhood aged 0-5 years in a separate paragraph.

Methods

Figure 1: It still looks blurry, is there a higher pixel version? Suggest the author to replace figure 1.

Results

Table 1, Page 15: Table 1 seems to lack an explanation of abbreviations, both in the title and in the contents of the table.

Table 2, Page 17: The information given in the table is on the low side, would it be considered to move the content to be accounted for in the body of the text instead of setting up a table?

Thank you and my best,

Your reviewer

Reviewer #3: Thank you for the opportunity to review PONE-D-24-06642R1.

The authors are commended for developing and evaluating a tool to assess ECEs knowledge, self-efficacy and attitudes towards physical literacy in early learning environments.

I have reviewed the article thoroughly, the previous reviewer’s comments and assessed the author’s revisions made to these comments.

General

Overall, I think that the manuscript could be greatly improved with more attention to the description of Physical Literacy (PL), and how this relates to and is distinct from Physical Activity (PL Consensus Statement, Tremblay et al., 2018). Despite the revisions, the manuscript would benefit from a stronger rationale for the scale and how this could be used as a proxy to evaluate PL interventions aimed at children. The originality of this study is not communicated. Greater detail and description of the holistic nature of the scale, the components involved and what this means would improve readability.

Finally, there remains too little discussion of how this study relates to others. For example, the discussion is very short and doesn’t capture all of the interesting elements of the data derived. The manuscript would be improved by providing more detailed discussion of similar studies to support the findings.

Specific

Abstract

Line 23-24 – I suggest improving this sentence by identifying “this construct”, as well as the type of interventions (i.e., PL or PA) you are referring to and how ECEs relate to PL among children.

Line 27 – What is meant by their “own personal context”?

Line 32 – The readability score of K-F Grade 8.7 was confusing until I read the manuscript. Consider adding more information here.

Line 41 – For clarity, I suggest including the type of intervention (e.g., PL among children).

Introduction

Line 68-69 – Bold statement “almost no data”. It is not clear what data does not exist.

The introduction reviews PA data from a wide range of populations rather than focusing on children in their early years. I suggest revising to concentrate on the population of focus.

The authors have done a great job including the definitions of PA and PL. It would be helpful if the reader had some specific examples of PL (e.g., specify fundamental movement skills) and how these two constructs support each other (e.g., “if you don’t learn to throw, then you won’t ever play baseball”) and also how they are distinct.

Line 110 – add “care” to end of sentence.

Methods

Why were cooks/chefs included as participants? This seemed odd.

Greater detail about how consent was obtained directly from the ECEs is important to include. The general statement at the end of this section does not effectively allow for replication or demonstrate procedures adequately.

Thank you again for the opportunity to review your article. I believe this is an important study that reports on the development and initial validation procedures for a scale that could be useful in the evaluation of future PL or PA intervention in the childcare environment. With some further revisions, the manuscript will be improved.

7. PLOS authors have the option to publish the peer review history of their article (what does this mean?). If published, this will include your full peer review and any attached files.

Reviewer #1: No

Reviewer #3: No

---

## [Author Response · Author response to Decision Letter 1]

24 Sep 2024

PONE-D-24-06642R1

Validity and reliability of a physical literacy knowledge, attitudes, self-efficacy and behaviors questionnaire for early childhood educators (PLKASB-ECE): An exploratory factor analysis

PLOS ONE

Journal Requirements #1

Any changes to the reference list should be mentioned in the rebuttal letter that accompanies your revised manuscript. 

Author Response (JR#1): We have removed the following references from the revised manuscript. In response to Reviewer #3 Comment #3 we significantly revised Lines 128-149 to strengthen rationale for the scale and explain the originality of the tool and study. This revision resulted in the removal of the following citations and the addition of new references. The new references cited are in red in the revised manuscript with track changes. 

• Brussoni et al. JMIR Res Protoc. 2021;10: e31041. doi:10.2196/31041

• McKinlay et al. Australas J Early Child. 2018;43: 32–42. doi:10.23965/AJEC.43.2.04

• Pölzl-Stefanec E. Br J Educ Technol. 2021;52: 2192–2208. doi:10.1111/bjet.13124

• Boldovskaia et al. PLOS ONE. 2023;18: e0288541. doi:10.1371/journal.pone.0288541

Editor Comment #1

I have one request. Can you make your headings the same font and font size as the other parts of the paper? 

Author Response (EC#1): We have selected all text and made all text the same font (Times New Roman) and font size (12 pt). We have verified that the headings are the same font and font size as all text in the manuscript.

Reviewer #1: ID: PONE-D-24-06642R1

Overall 

Reviewer #1 Comment #1: The author has made careful revisions and the quality of the article has been greatly improved, congratulations! However, some detailed issues still need to be improved.

Author Response (R1-C1): Thank you for the opportunity to revise this manuscript. We appreciate all the feedback and feel that the changes have improved the work. Please see our changes based on the additional suggestions.

Abstract

Reviewer #1 Comment #2: Overall, Subheadings can be used to separate the content of the abstract, including “Background”, “Objective”, “Methods”, “Results”, “Conclusion”, etc.

Author Response (R1-C2): Thank you for this suggestion. We have updated the abstract to include each subheading.

Introduction

Reviewer #1 Comment #3: Line 66-74, Page 4: Perhaps it would make more sense to present the situation of the group of early childhood aged 0-5 years in a separate paragraph. 

Author Response (R1-C3): We have separated the paragraph to present the context of early childhood 0-5 year on its own (starting at Line 71). 

Methods

Reviewer #1 Comment #4: Figure 1: It still looks blurry, is there a higher pixel version? Suggest the author to replace figure 1. 

Author Response (R1-C4): Thank you for mentioning this. We have resubmitted the image at a higher resolution (300 dpi) 2250 × 1522 according to journal instructions.

Results

Reviewer #1 Comment #5: Table 1, Page 15: Table 1 seems to lack an explanation of abbreviations, both in the title and in the contents of the table.

Author Response (R1-C5): A notes section has been added to provide explanation of abbreviations in table 1 (Line 329 to 332).

Reviewer #1 Comment #6: Table 2, Page 17: The information given in the table is on the low side, would it be considered to move the content to be accounted for in the body of the text instead of setting up a table?

Author Response (R1-C6): While we could provide this information in the body of the text, having the information in a table makes the comparison between the four-factor model and the five factor model a quick read that provides the reader with information quickly. We suggest that keeping it in a table is a reader friendly decision and have left it as a table. 

Reviewer #3: ID: PONE-D-24-06642R1

Overall 

Reviewer #3 Comment #1: The authors are commended for developing and evaluating a tool to assess ECEs knowledge, self-efficacy and attitudes towards physical literacy in early learning environments.

Author Response (R3-C1): We appreciate the feedback.

Reviewer #3 Comment #2: Overall, I think that the manuscript could be greatly improved with more attention to the description of Physical Literacy (PL), and how this relates to and is distinct from Physical Activity (PL Consensus Statement, Tremblay et al., 2018). 

Author Response (R3-C2): We have revised the introduction to provide a clearer distinction between PL and PA. This now reads. 

“Physical literacy encompasses a comprehensive approach to increasing physical activity participation which in turn supports child development [16]. While physical activity relates to bodily movement, Canada’s Physical Literacy Consensus Statement defines physical literacy as “the motivation, confidence, physical competence, knowledge, and understanding to value and take responsibility for engagement in physical activities throughout life” [17]. It addresses the lifelong journey of who, what, where, when, why, and how to cultivate the engagement in physical activity that enhances personal health. Physical literacy can be conceptualized in a framework of four interrelated domains: affective, physical, cognitive, and behavioral [17]. This holistic approach emphasizes the social processes associated with lifelong learning and sustained physical activity engagement [17–19].” 

Reviewer #3 Comment #3: Despite the revisions, the manuscript would benefit from a stronger rationale for the scale and how this could be used as a proxy to evaluate PL interventions aimed at children. The originality of this study is not communicated. Greater detail and description of the holistic nature of the scale, the components involved and what this means would improve readability. 

Author Response (R3-C3): Thank you for this comment. We have significantly revised Lines 128-149 to strengthen rationale for the scale and explain the originality of the tool and study. 

Reviewer #3 Comment #4: Finally, there remains too little discussion of how this study relates to others. For example, the discussion is very short and doesn’t capture all of the interesting elements of the data derived. The manuscript would be improved by providing more detailed discussion of similar studies to support the findings. 

Author Response (R3-C4): We appreciate this comment. We have expanded the discussion section Lines 435 to 445 to capture some additional elements of the data derived. We have added information to the discussion section to provide context of our study to similar studies. See Lines 448 to 490.

Abstract

Reviewer #3 Comment #5: Line 23-24 – I suggest improving this sentence by identifying “this construct”, as well as the type of interventions (i.e., PL or PA) you are referring to and how ECEs relate to PL among children. 

Author Response (R3-C5): We have revised the background section of the abstract (Lines 23-25) to address these three suggestions to “However, enhancing ECEs’ ability to foster physical literacy requires valid and reliable assessment tools to measure holistic physical literacy constructs (i.e., cognitive, affective, behavioral) in order to assess the impact of physical literacy educational interventions.”

Reviewer #3 Comment #6: Line 27 – What is meant by their “own personal context”? 

Author Response (R3-C6): We are referring to the educator’s personal relationship with movement. This is relevant as personal affinity for movement has been found to be a predictor of providing more physical activity opportunities for children which is discussed in more detail in the Introduction paragraph 5. To improve the abstract, we have revised Line 29 to “the educator’s own personal physical literacy behaviours.”

Reviewer #3 Comment #7: Line 32 – The readability score of K-F Grade 8.7 was confusing until I read the manuscript. Consider adding more information here.

Author Response (R3-C7): Thank you for the suggestion. We have edited the sentence for clearer language, and provided the full name "Flesch-Kincaid Grade Level" as "K-F" might not be immediately recognizable to all readers. This sentence now reads “Six additional items were added, resulting in a final 25-item self-report measure with a readability score equivalent to an 8th grade reading level (Flesch-Kincaid Grade Level: 8.7).”

Reviewer #3 Comment #8: Line 41 – For clarity, I suggest including the type of intervention (e.g., PL among children).

Author Response (R3-C8): We have improved the clarity of the sentence to include the type of intervention (Line 34-35). The sentence now reads “This tool will be valuable as a research outcome measure for assessing educational interventions aimed at enhancing educators’ understanding, confidence, and strategies for fostering holistic concepts of physical literacy in young children within ECECs.”

Methods

Reviewer #3 Comment #9: Why were cooks/chefs included as participants? This seemed odd.

Author Response (R3-C9): In Table 1 we include the heading Position where Cook/Chef appears. In ECECs within the province of Alberta, all individuals who can be part of the education and care of the ECEC children must have a minimum of Level 1 education. Some Cook/Chefs have completed the Level 1 educator training and work with children when they are not working preparing meals. This means they engage with children directly in the various components of the day which includes aspects of physical activity (indoor/outdoor time). In this way they impact the physical literacy environment and need to be part of the population on which the tool has been validated. Not all centers have Cook/Chefs, not all Cook/Chefs have completed the Level 1 education, so the relatively low percentage of Cook/Chefs is representative of the provincial population. We hope that contextualizes cook/chef within this participant group.

Reviewer #3 Comment #10: Greater detail about how consent was obtained directly from the ECEs is important to include. The general statement at the end of this section does not effectively allow for replication or demonstrate procedures adequately.

Author Response (R3-C10): Thank you for pointing this out. We have provided more detail in the Methods section (Line 214-218) which now reads “Centers that agreed to participate invited three educators from their center to join the research study. Each educator received a personal email containing: information about study participation, private secure digital access to their consent form, and contact information for a trained research associate to answer potential questions.”

---

## [Editor Report · Decision Letter 2]

6 Oct 2024

PONE-D-24-06642R2Validity and reliability of a physical literacy knowledge, attitudes, self-efficacy and behaviors questionnaire for early childhood educators (PLKASB-ECE): An exploratory factor analysisPLOS ONE

Dear Dr. Lafave,

Thank you for submitting your manuscript to PLOS ONE. After careful consideration, we feel that it has merit but does not fully meet PLOS ONE’s publication criteria as it currently stands. Therefore, we invite you to submit a revised version of the manuscript that addresses the points raised during the review process.

Please include the following items when submitting your revised manuscript:A rebuttal letter that responds to each point raised by the academic editor and reviewer(s). You should upload this letter as a separate file labeled 'Response to Reviewers'.A marked-up copy of your manuscript that highlights changes made to the original version. You should upload this as a separate file labeled 'Revised Manuscript with Track Changes'.An unmarked version of your revised paper without tracked changes. You should upload this as a separate file labeled 'Manuscript'.If applicable, we recommend that you deposit your laboratory protocols in protocols.io to enhance the reproducibility of your results. Protocols.io assigns your protocol its own identifier (DOI) so that it can be cited independently in the future. For instructions see: https://journals.plos.org/plosone/s/submission-guidelines#loc-laboratory-protocols. Additionally, PLOS ONE offers an option for publishing peer-reviewed Lab Protocol articles, which describe protocols hosted on protocols.io. Read more information on sharing protocols at https://plos.org/protocols?utm_medium=editorial-email&utm_source=authorletters&utm_campaign=protocols.

We look forward to receiving your revised manuscript.

Kind regards,

Mary Diane Clark, PhD

Academic Editor

PLOS ONE

Journal Requirements:

Additional Editor Comments:

Thank you for all of the work on this mansucript. It is clearer and much easier to read. There is not a copy editor for the journal so I have identified a few very minor issues that will take you just a few minutes to correct. Thank yo very much

---

## [Author Response · Author response to Decision Letter 2]

8 Oct 2024

PONE-D-24-06642R2

Validity and reliability of a physical literacy knowledge, attitudes, self-efficacy and behaviors questionnaire for early childhood educators (PLKASB-ECE): An exploratory factor analysis

PLOS ONE

Journal Requirements 

Journal Requirements #1

Any changes to the reference list should be mentioned in the rebuttal letter that accompanies your revised manuscript. 

Author Response #1: We have removed the following references from the revised manuscript in response to Editor Comment #3.

32. Early childhood grows up: Towards a critical ecology of the profession: Setting the scene. Early Childhood Grows Up: Towards a Critical Ecology of the Profession. 2012. pp. 3–19. doi:10.1007/978-94-007-2718-2_1

33. Ajzen I. The theory of planned behavior. Theor Cogn Self-Regul. 1991;50: 179–211. doi:10.1016/0749-5978(91)90020-T

36. Janz NK, Becker MH. The Health Belief Model: A Decade Later. Health Educ Q. 1984;11: 1–47. doi:10.1177/109019818401100101

37. Ajzen I, Fishbein M. Attitudes and the Attitude-Behavior Relation: Reasoned and Automatic Processes. Eur Rev Soc Psychol. 2000;11: 1–33. doi:10.1080/14792779943000116

38. Bandura A. Social learning theory. Englewood Cliffs, N.J: Prentice Hall; 1977. 

Editor

Editors - Thank you for all of the work on this mansucript. It is clearer and much easier to read. There is not a copy editor for the journal so I have identified a few very minor issues that will take you just a few minutes to correct. Thank yo very much

Authors – thank you very much for the opportunity and effort to make this a better paper. We are grateful for the review and comments.

Editor Comment #1 - Objective: The purpose of this study was to develop a holistic digital tool to measure physical literacy knowledge, attitudes, self-efficacy, and behaviors of early childhood educators (PLKASB-ECE) for both their professional teaching context as well as the educator’s own

Author Response #1 – Line #27 reflects this change

Editor Comment #2 - A 78 systematic review of objectively measured physical activity in center-based child care revealed that 79 physical inactivity levels are high across worldwide geographical locations [13] 

Author Response #2 – Line #78 reflects this change

Editor Comment #3 – This section seems like a list of statements that is not supporting the paragraph it is in—please take it out

“Exploring educator’s social cognitions can provide deeper insights 117 into how they can influence their behaviors in supporting the development of children’s physical 118 literacy. A complex set of factors govern educational decisions and behaviors within the ECEC 119 environment. The critical ecology framework [32] proposes that educator behaviors are 120 influenced by relationships and social interactions within the various contexts of an ECEC 121 environment. Social cognitive factors have been used to understand why people engage in certain 122 behaviors and are believed to predict intentional behavior [33–36]. Social cognitions include 123 constructs such as attitude and efficacy beliefs. Attitude is defined as the perceived favorable or 124 unfavorable evaluation of a certain behavior [37], while self-efficacy refers to the belief in one’s 125 ability to perform a particular behavior [38].”

Author Response #3 – Line #116-125 have been removed. The reference list will reflect the deletion of these citations. Deleted citations include 32, 33, 36, 37, and 38.

Editor Comment #4 – Please make these into three separate sentences 

“Six were review articles [63–68], twelve addressed kindergarten to grade 6 [69–76] or 286 grade 7 to 12 [18,77–79] school aged children motor skills measurement of physical literacy, one 287 article described the development of a physical literacy questionnaire for primary school teachers 14 288 [80], and one provided a qualitative assessment of physical literacy in primary school teachers 289 [81].” 

Author Response #4 – This section has been revised. It now reads –

“Six review articles [63–68] examined literature on physical literacy. Twelve articles focused on the measurement of motor skills related to physical literacy in school-aged children, specifically from kindergarten to grade 6 [69–76] and grades 7 to 12 [18,77–79]. Additionally, one article described the development of a physical literacy questionnaire for primary school teachers [80], and another provided a qualitative assessment of physical literacy among this group [81].”

Editor Comment #5 – This sentence is extremely long---I think it would be easier to read if they were separate sentences 

“It consists of six subscales: 1) Knowledge sum score (n=7) 388 measuring educators’ physical literacy knowledge; 2) Perception of Knowledge (n=2) measuring 389 educators’ perception of their understanding related to physical literacy concepts; 3) 390 Attitude (n=3) measuring educators’ attitudes regarding physical literacy and health outcomes; 4) 391 Self Efficacy (n=3) measuring educator’s self-efficacy towards developing physical literacy 392 skills in the early learning environment; 5) Environment Behaviors (n=3) measuring educators’ 3 393 perception of their physical literacy promotion practices in the ECE environment; 6) 394 and Personal Behaviors (n=3) measuring perception of personal physical activity behaviors 395 (Table 6).” 

Author Response #5 – This section has been revised. It now reads –

The instrument consists of six subscales. The first is the Knowledge Sum Score (n=7) measuring educators’ physical literacy knowledge by summing all correct answers for the score. The second subscale is the Perception of Knowledge (n=2) measuring educators’ perception of their understanding related to physical literacy concepts. The third subscale is Attitude (n=3) measuring educators’ attitudes regarding physical literacy and health outcomes. The fourth subscale is Self-Efficacy (n=3) measuring educator’s self-efficacy towards developing physical literacy skills in the early learning environment. The fifth subscale is Environment Behaviors (n=3) measuring educators’ perception of their physical literacy promotion practices in the ECE environment. The sixth subscale is Personal Behaviors (n=3) measuring perception of personal physical activity behaviors (Table 6).

Editor Comment #6 –they relate to health [17]. The PLKASB-ECE subscales of knowledge and perception of knowledge address within this

Author Response #6 – Line #467 reflects this request. I have also changed the subscale capitalization in this paragraph to align with 388-397.

The PLKASB-ECE subscales of Knowledge and Perception of Knowledge address within this domain. The affective domain refers to the motivation and confidence to incorporate movement into daily life [17]. The PLKASB-ECE subscales of Attitude and Self-Efficacy address belief that physical activity supports health as well as confidence in being able to include it within daily practice. As such these subscales fit within the affective domain. The behavioral domain addresses the engagement in physical activities on a daily basis [17]. The PLKASB-ECE subscales of Environmental Behavior and Personal Behavior speak to the educator actions related to taking responsibility for physical literacy experiences with their classroom setting as well as including physical activity in their personal daily practice which align with the behavioural domain.

Editor Comment #7 - 

473 competence required to participate in various activities [17]. We identified existing tools in the 474 literature targeting fundamental skill development for preschool (3-5 years) children that relate to 4 475 this physical competence [59–62] as this aspect of physical literacy was not included in the 476 PLKASB-ECE.

Author Response #7 – Line #478 reflects this change

Editor Comment #8 - 

In the Strength and Limitations sections please take about the numbers

Author Response #8 – I believe this request pertains to the removal of numbers regarding the proportion of centers and women. These numbers have been removed from Line 510-515 and now reads: 

The proportion of recruited centers (large urban, medium, and small population centers) were relatively similar to overall population distribution across the province [50]. Some limitations should be considered when interpreting the findings of this study. First, the study sample population was limited to the province of Alberta, Canada which limits the generalizability of the findings. Second, most participants were women. However, this proportion is similar to the ECEC workforce in Canada so this sampling is reflective of the intended population [91].

---

## [Editor Report · Decision Letter 3]

14 Oct 2024

Validity and reliability of a physical literacy knowledge, attitudes, self-efficacy and behaviors questionnaire for early childhood educators (PLKASB-ECE): An exploratory factor analysis

PONE-D-24-06642R3

Dear Dr. Lafave,

We’re pleased to inform you that your manuscript has been judged scientifically suitable for publication and will be formally accepted for publication once it meets all outstanding technical requirements.

Kind regards,

Mary Diane Clark, PhD

Academic Editor

PLOS ONE

Additional Editor Comments (optional):

Thank you very much for all of th collaboration on the paper. I am recommending acceptance.
---

## [Editor Report · Acceptance letter]

18 Oct 2024

PONE-D-24-06642R3 

PLOS ONE

Dear Dr. Lafave, 

I'm pleased to inform you that your manuscript has been deemed suitable for publication in PLOS ONE. Congratulations! Your manuscript is now being handed over to our production team.

Kind regards, 

on behalf of

Dr. Mary Diane Clark 

Academic Editor

PLOS ONE